# Linear Maps, Contrastive Objectives: A Principled Strategy for fMRI Decoding Consistent Across Modalities

## Abstract

A prominent theory in cognitive science suggests that concepts in the brain are organized as high-dimensional vectors, with semantic meaning captured by directions and relative angles in this space. Brain decoding is the effort of reconstructing or retrieving stimuli (or their representations) from neural activity and involves finding a function that approximates how the brain represents concepts. This motivates the investigation of contrastive objectives as biologically plausible candidates to reverse the brain loss function. In this work, we study how functional MRI (fMRI) activity can generally be aligned with the embedding spaces of foundation models in vision, language, and audio. Although neural computations are highly non-linear at the microscale, fMRI measurements average signals across space and time, further smoothed by noise, effectively linearizing the observable representation. Consistent with these views, our experiments across multiple datasets demonstrate that linear contrastive decoders consistently outperform ridge regression and standard non-linear alternatives, and that these results generalize across images, text, and sound. These findings indicate that decoding gains arise more from the choice of training objective than from architectural complexity, pointing to contrastive-linear models as a principled strategy for brain decoding.

## 1 Introduction

A central challenge in cognitive science is to understand how the brain represents concepts and encodes sensory information. Recent theoretical work argues that human concepts are most plausibly represented as high-dimensional vectors (Piantadosi & et al., 2024). Vector-based representations naturally explain typicality and similarity effects through distances in the representational space, capture relations and analogies via vector arithmetic, support compositionality and theory-like structures, and even allow the flexible formation of ad hoc categories. This framework unifies different theory-based views of concepts under a single representational format. Moreover, it resonates with the success of modern foundation models, which learn rich embedding spaces where meaning is encoded in geometric relations among vectors. If the brain organizes concepts in such vector spaces, then comparison and learning are likely driven by similarity, suggesting that contrastive learning provides a biologically plausible approximation of the brain's own optimization principle.

At the same time, a complementary line of research has challenged the assumption that modeling brain dynamics at the macroscale necessarily requires complex non-linear systems. A recent large-scale study from Nozari et al. (2024) on fMRI data showed that linear models not only match but often outperform a wide range of non-linear approaches across predictive accuracy, residual structure, and computational efficiency. The authors traced this apparent linearity to several factors intrinsic to macroscopic measurements: spatial averaging over millions of neurons, temporal filtering of fast dynamics, observation noise, and the limited sample size relative to dimensionality. Together, these effects act to smooth and linearize the measured signal, such that what fMRI captures is effectively a first-order approximation of the underlying neural computations. This provides a principled explanation for why linear models can be highly effective in fMRI decoding, despite the non-linear nature of the neural processes they ultimately reflect.

Building on these perspectives, this work is motivated by several key questions: What is the most effective way to map brain activity into the embedding spaces of foundation models? Do more complex non-linear models provide an advantage in the context of noisy and high-dimensional data? Is decoding performance driven more by vector alignment in the representational space than by average error minimization?

To address these questions, we systematically study brain decoding from functional Magnetic Resonance Imaging (fMRI) across three distinct modalities (images, music, and text), using embeddings extracted from state-of-the-art foundation models. We evaluate a spectrum of decoding models, ranging from ridge regression to linear mappings trained with contrastive loss, to shallow MLPs. Our findings reveal three key insights: (i) a simple linear mapping trained with contrastive learning consistently outperforms ridge regression across modalities; (ii) introducing non-linearities via MLPs does not improve decoding performance, and in fact degrades retrieval accuracy; (iii) prioritizing discriminative separation of embeddings is more important than minimizing pointwise error.

We emphasize that our conclusion is related to the data regime and preprocessing commonly used in fMRI decoding (GLM betas or HRF-averaged responses). We do not claim that non-linear architectures cannot outperform linear ones in other settings—for example, with minimally averaged data, larger datasets, or temporal models.

## 2 RELATED WORKS

Recent years have witnessed remarkable progress in decoding complex stimuli from neural activity, particularly in non-invasive settings such as fMRI (Gallant et al., 2012; Huth et al., 2016; Ferrante et al., 2024a; Banville et al., 2025). In the visual domain, approaches leveraging pre-trained vision–language models such as CLIP, combined with linear regression or contrastive learning, have enabled retrieval-based decoding and even realistic image reconstruction when coupled with diffusion models (Ozcelik & VanRullen, 2023; Lin et al., 2022; Scotti et al., 2023; Chen et al., 2023a; Xia et al., 2024a). Beyond vision, growing evidence shows that fMRI activity can be mapped onto latent spaces of diverse modalities—including video (Chen et al., 2023b), language and music (Tang et al., 2023; Jalouzot et al., 2025; Denk et al., 2023; Ferrante et al., 2024b). The advent of large pre-trained models has been a key enabler of this progress, providing rich representational spaces that support both retrieval tasks and generative reconstruction from neural data.

Brain decoding approaches have traditionally relied on linear methods such as ridge regression to predict high-dimensional representations of stimuli from non-invasive neural recordings (Ozcelik & VanRullen, 2023; Liu et al., 2023; Denk et al., 2023). While successful in controlled settings, these models are limited in their ability to capture the semantic richness of natural stimuli. More complex neural networks, in contrast, have shown promising results (Scotti et al., 2024; Xia et al., 2024b; Careil et al., 2025) but often at the cost of overfitting or reduced interpretability, leaving unclear whether performance gains come from non-linear modeling or from task-specific processing. Furthermore, most studies have focused on a single modality (e.g., vision or language), leaving open the question of whether decoding strategies generalize across different types of cognitive data.

## 3 MATERIAL & METHODS

An overview of the proposed framework is shown in Figure 1, while the model architecture is described in more detail in Section 3.4. Our approach aims to learn a shared representational space in which neural responses and stimulus embeddings can be directly compared. Once trained, the model enables retrieval of the corresponding stimulus representation (text, image, or audio) from neural activity alone. The experiments are conducted independently for each stimulus modality, using three distinct datasets. Importantly, the same model architecture is employed across all modalities, demonstrating that our framework achieves consistent improvements over baseline methods and more complex non-linear models. The following subsections provide details on the datasets, the decoder design, the training objective, and the evaluation metrics.

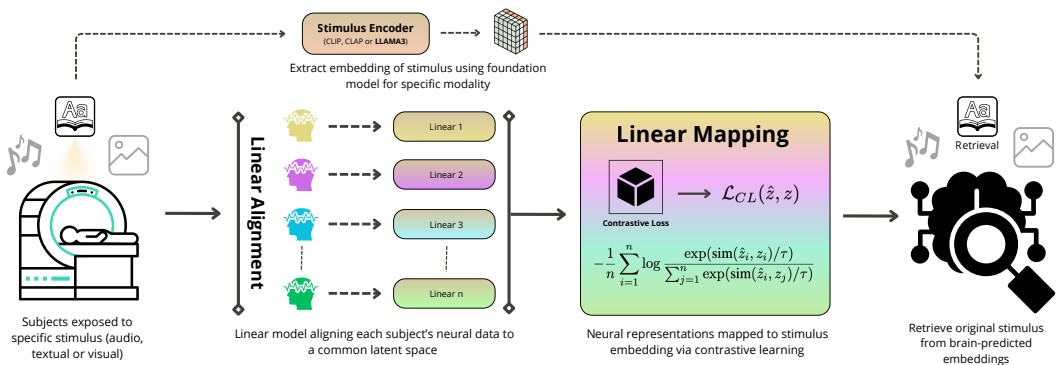

Figure 1: The same linear contrastive model is employed across three experimental conditions, differing only in the stimulus modality (audio, textual, or visual). For each modality, neural responses from fMRI are aligned through subject-specific linear transformations and mapped into the corresponding stimulus embedding space (obtained from a pretrained foundation model such as CLIP for images, CLAP for audios, or LLaMA for text) via a contrastive learning objective. Training is carried out independently for each modality. At test time, retrieval is performed by comparing brain-predicted embeddings with estimated stimulus embeddings.

## 3.1 IMAGE PROCESSING

For the visual dataset, we relied on the *Natural Scenes Dataset* (NSD) from Allen et al. (2022), which includes fMRI data acquired while multiple subjects viewed natural images. The dataset provides a large number of fMRI trials (over 24,000) and two distinct subsets: a training set of approximately 8,859 unique images per subject, and (ii) a common set of 982 images viewed by all subjects, used for alignment and testing. In order to reduce dimensionality, the fMRI signals were preprocessed by applying NSD General ROI masks and estimating beta coefficients through a general linear model (GLM) (Kay et al., 2013; Prince et al., 2022), which included a fitted hemodynamic response function (HRF) correction and a denoising process. We focused on data from Subj01, Subj02, Subj05, Subj07 then normalized and transformed into the MNI space at 2 mm resolution, reducing computational cost and enabling inter-subject comparison. Following common practice in all major NSD decoding papers (Takagi & Nishimoto, 2023), we retain the most reliable voxels in V1–V4. Each neural sample is represented as a vector of 15,724 voxels, corresponding to the estimated fMRI beta responses within selected visual ROIs.

For each stimulus, the corresponding natural image is fed into CLIP model (Radford et al., 2021) at inference mode, obtaining its high-level semantic representation given by a 512-dimensional embedding (image-text projection layer).

## 3.2 LANGUAGE PROCESSING

We used the publicly available dataset introduced in LeBel et al. (2023), focusing on three subjects (S1, S2, and S3). In the manuscript, we will refer to this dataset as HUTH Language. Each subject underwent approximately 16 hours of fMRI recordings while listening to 83 naturalistic stories taken from the *The Moth* and *Modern Love* podcasts. fMRI data were acquired with a 3T Siemens Skyra scanner using a repetition time (TR) of 2.00 s and an isotropic voxel size of 2.6 mm. Standard preprocessing included motion correction, cross-run alignment, standardization, and removal of low-frequency drifts. For training, we used the first 70 stories of each subject, reserving 12 for validation. Additionally, the story *wheretheressmoke* (consisting of 250 sentences) was presented 10 times to improve the signal-to-noise ratio in the test set.

To reduce complexity and restrict our analysis to language-sensitive regions, we used an encoding model that mapped word embeddings from a large language model (Dubey et al., 2024) (LLaMA3-8B, layer 13) to fMRI responses. Word embeddings were computed in context windows of five preceding words and then downsampled with a Lanczos filter to match the temporal resolution of the fMRI signal. We follow prior work on voxel filtering in language decoding (Tang et al., 2023; Toneva

& Wehbe, 2019): we select voxels with an encoding model mapping from stimulus embeddings to fMRI responses and keep the ones with higher predictive correlation on held-out data. Pearson correlation between predicted and observed activity was calculated, and the top 10,000 cortical voxels showing the highest predictability were selected as target regions for decoding.

## 3.3 MUSIC PROCESSING

We employed the GTZan fMRI dataset Nakai et al. (2022), which consists of recordings from five participants (sub-001 to sub-005) who each listened to 540 music excerpts evenly distributed across ten genres (blues, classical, country, disco, hip-hop, jazz, metal, pop, reggae, and rock). Each stimulus lasted 15 seconds with a 2-second fade-in and fade-out, sampled at 22.05 kHz. The experimental design included 18 runs per subject (12 for training, 6 for testing), each composed of 40 clips. fMRI data were acquired with a 3.0T scanner at TR = 1.5 s (400 volumes per run). Preprocessing included motion correction, co-registration to the MNI template using FSL, detrending with Nilearn to remove low-frequency drifts, run-wise standardization, and a hemodynamic delay correction (discarding the first 3 TRs, i.e., 4.5 s). Neural responses for each stimulus were then averaged over the following 10 TRs (15 s), yielding one fMRI representation per excerpt. The resulting dataset comprised 480 training pairs and 60 test pairs per subject.

For the data-driven voxel selection, we constructed voxel-wise regression models to predict brain activity from CLAP (Elizalde et al., 2023) latent music embeddings (512-dimension from audio-text projection layer). Each voxel was modeled independently, with regularization hyperparameter $\alpha$ optimized through nested cross-validation. Prediction quality was assessed by the Pearson correlation between predicted and observed responses on held-out training data. Voxels surpassing a correlation threshold were retained (above 3,000), yielding masks of music-responsive regions of interest. These voxels served as the input space for subsequent alignment and decoding analyses. See Ferrante et al. (2024b) for details.

## 3.4 NEURAL-TO-EMBEDDING DECODER

We designed a neural architecture to learn a mapping between neural activity and the target embedding space using a contrastive learning framework. The best architecture is simple: the decoder is composed of a sequence of linear layers, so that the transformation is essentially a stack of affine projections.

Formally, given an input vector $\mathbf{x} \in \mathbb{R}^d$ and a subject index $k$, the model first applies a subject-specific alignment layer $A_k$, followed by hidden projections and an output layer: $\mathbf{z} = W_o W_h A_k \mathbf{x}$, where $A_k \in \mathbb{R}^{d \times d_c}$ aligns the subject-specific input to a shared dimensionality $d_c$, $W_h \in \mathbb{R}^{d_c \times h}$ projects to hidden dimension $h$, and $W_o \in \mathbb{R}^{h \times d_o}$ maps to the output embedding space of foundation model $d_o$. See Appendix for details about subject alignment.

In order to align neural representations with target embeddings $\mathbf{y} \in \mathbb{R}^{d_o}$, we employ a contrastive loss inspired by the NT-Xent formulation. For a batch of predicted embeddings $\{\mathbf{z}_i\}_{i=1}^N$ and targets $\{\mathbf{y}_i\}_{i=1}^N$, the cosine similarity is computed as $S_{ij} = \frac{\mathbf{z}_i^\top \mathbf{y}_j}{\|\mathbf{z}_i\|\|\mathbf{y}_j\|}$. The loss encourages each $\mathbf{z}_i$ to be most similar to its paired $\mathbf{y}_i$:

$$\mathcal{L}_{\text{contrastive}} = -\frac{1}{N} \sum_{i=1}^N \log \frac{\exp(S_{ii}/\tau)}{\sum_{j=1}^N \exp(S_{ij}/\tau)},$$

where $\tau > 0$ is a temperature hyperparameter.

The decoder is trained end-to-end with AdamW optimization, using early stopping based on validation loss. As references, we tested (i) a ridge regression model mapping neural representations directly to the embedding space, and (ii) an MLP decoder with non-linear activations between layers.

## 3.5 EVALUATION

At test time, we run the decoder in inference mode to obtain predicted embeddings. Given a batch $\{(\mathbf{x}_i, \mathbf{y}_i, k_i)\}_{i=1}^N$, where $k_i$ is the subject index, we compute $\hat{\mathbf{y}}_i = f_\theta(\mathbf{x}_i; k_i)$, and collect all predictions $\{\hat{\mathbf{y}}\}$ and corresponding ground-truth targets $\{\mathbf{y}_i\}$ for retrieval-based evaluation.

Evaluation is performed within subject to factor out inter-subject variability. For each subject $s$, we have a *query set* $\mathcal{Q}_s = \{\hat{\mathbf{y}}_i^{(s)}\}_{i=1}^{n_s}$ (predicted embeddings) and a *reference set* $\mathcal{R}_s = \{\mathbf{y}_j^{(s)}\}_{j=1}^{n_s}$ (ground-truth embeddings). Each query $\hat{\mathbf{y}}_i^{(s)}$ has a unique paired target $\mathbf{y}_i^{(s)}$ in the reference set. Correlation is measured with the cosine similarity, and the corresponding cosine distance (to be minimized) is $d_{\cos}(\hat{\mathbf{y}}, \mathbf{y}) = 1 - \cos(\hat{\mathbf{y}}, \mathbf{y})$.

For each subject $s$, we perform a nearest-neighbor search within $\mathcal{R}_s$ using cosine distance. Concretely, for each query $\hat{\mathbf{y}}_i^{(s)}$ we compute all pairwise distances $d_{ij}^{(s)} = d_{\cos}\left(\hat{\mathbf{y}}_i^{(s)}, \mathbf{y}_j^{(s)}\right)$, $j = 1, \ldots, n_s$, rank reference embeddings by ascending distance, and select the $k$ closest matches: $\Pi_i^{(s)}(k) = \{j_1, \ldots, j_k\}$ with $d_{ij_1}^{(s)} \leq \cdots \leq d_{ij_k}^{(s)}$.

Let $j^\star = i$ denote the index of the correct target for query $i$. The Top-$k$ accuracy for subject $s$ is defined as

$$\text{Top-}k(s) = \frac{1}{n_s} \sum_{i=1}^{n_s} \mathbb{1}\big[\, j^\star \in \Pi_i^{(s)}(k) \,\big],$$

i.e., the fraction of queries for which the true target appears among the $k$ nearest neighbors. Overall performance is reported as the micro-average across all subjects:

$$\text{Top-}k = \frac{\sum_s \sum_{i=1}^{n_s} \mathbb{1}\big[\, j^\star \in \Pi_i^{(s)}(k) \,\big]}{\sum_s n_s}.$$

In particular, we report Top-1 and Top-3 accuracies for each stimulus modality. Given the sample size of the test set the chance level is Top-1: 1/250=0.40% and Top-3: 3/250=1.20% for HUTH dataset, Top-1: 1/980=0.10% and Top-3: 3/980=0.31% for NSD, Top-1: 1/60=1.67% and Top-3: 3/60=5.00% for GTZAN.

## 4  RESULTS

Table 1: Retrieval accuracies (mean $\pm$ std) per dataset/metric.

| Dataset | Metric | Method | Accuracy (%) |
|---------|--------|--------|--------------|
| NSD (Image) | Top-1 | Ridge Reg. | $15.79 \pm 0.89$ |
| | | Linear CL | $21.80 \pm 0.76$ |
| | | Non-Linear CL | $17.76 \pm 1.71$ |
| | Top-3 | Ridge Reg. | $29.53 \pm 1.57$ |
| | | Linear CL | $39.66 \pm 0.91$ |
| | | Non-Linear CL | $35.20 \pm 1.33$ |
| HUTH (Language) | Top-1 | Ridge Reg. | $29.11 \pm 3.23$ |
| | | Linear CL | $42.04 \pm 2.19$ |
| | | Non-Linear CL | $38.23 \pm 2.68$ |
| | Top-3 | Ridge Reg. | $51.33 \pm 2.24$ |
| | | Linear CL | $66.25 \pm 2.87$ |
| | | Non-Linear CL | $61.70 \pm 1.83$ |
| GTZAN (Music) | Top-1 | Ridge Reg. | $22.67 \pm 1.56$ |
| | | Linear CL | $33.13 \pm 1.47$ |
| | | Non-Linear CL | $25.39 \pm 1.11$ |
| | Top-3 | Ridge Reg. | $49.10 \pm 2.00$ |
| | | Linear CL | $57.97 \pm 1.12$ |
| | | Non-Linear CL | $52.30 \pm 1.60$ |

Across all three modalities, decoding performance shows a consistent advantage for the linear contrastive model (Table 1 & Figure 3). In the visual domain, linear contrastive learning achieved the highest retrieval accuracies, clearly outperforming both ridge regression and the non-linear variant.

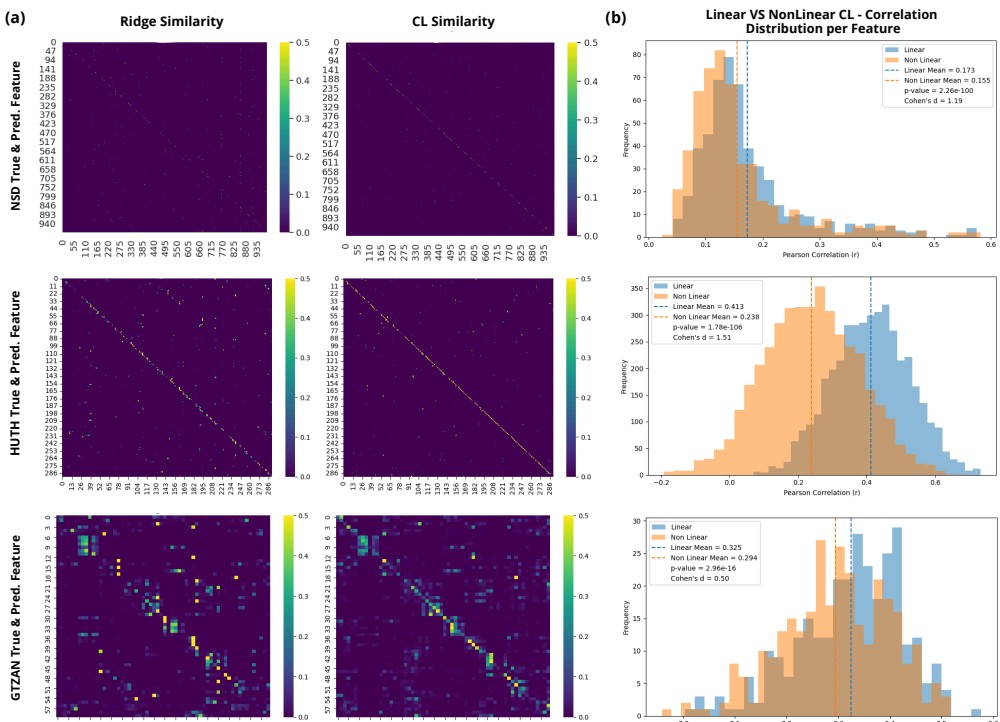

Figure 2: **(a)** Each heatmap represents the cosine similarity between predicted and ground-truth stimulus embeddings, computed per pair of features. Results are shown for three datasets (NSD, HUTH, GTZAN) and two models: a linear Ridge regression (left column) and the best contrastive learning model (the linear one, right column). The diagonal reflects correct predictions with high similarity between corresponding stimuli, while off-diagonal values indicate confusion between different candidates. All similarity matrices are normalized using a row-wise softmax to emphasize alignment between prediction and target embeddings. The CL model produces more concentrated diagonal patterns, indicating superior matching accuracy compared to Ridge approach. **(b)** Each plot displays the distribution of Pearson correlation coefficients computed between model predictions (Linear in blue, Non-Linear in orange) and the ground truth stimulus embeddings, evaluated separately for each embedding feature. Dashed vertical lines indicate the mean correlation for each model. A t-test was performed for each comparison, testing the alternative hypothesis that the linear model yields higher correlations than the non-linear model. The resulting p-value and effect size (Cohen's d) are reported in the legend. Results demonstrate that the linear model consistently achieves significantly higher correlations, with effect sizes ranging from moderate (Cohen's d = 0.50) to large (Cohen's d > 1), depending on the dataset.

A similar trend was observed in the language domain, where the linear contrastive model provided the largest gains, with improvements particularly pronounced at the Top-3 level. In the musical domain, the same model again yielded superior accuracies, indicating that the benefits of contrastive learning extend beyond a single modality. This improvement is also qualitatively evident in Figure 2 (left panel), where correct embedding pairs (real and brain-predicted) are marked by sharper diagonal activations in the similarity matrices.

Comparisons between linear and non-linear mappings further demonstrate that architectural complexity does not translate into performance gains. Despite introducing additional parameters and activation functions (Table A5 in Appendix) between subject aligner and mapping layer, non-linear models consistently underperformed compared to the linear contrastive approach. Feature-wise evaluation (Figure 2, right panel) confirmed that linear mappings lead to stronger correlations between predicted and ground-truth embeddings, suggesting that the critical factor is the contrastive objective itself rather than model complexity.

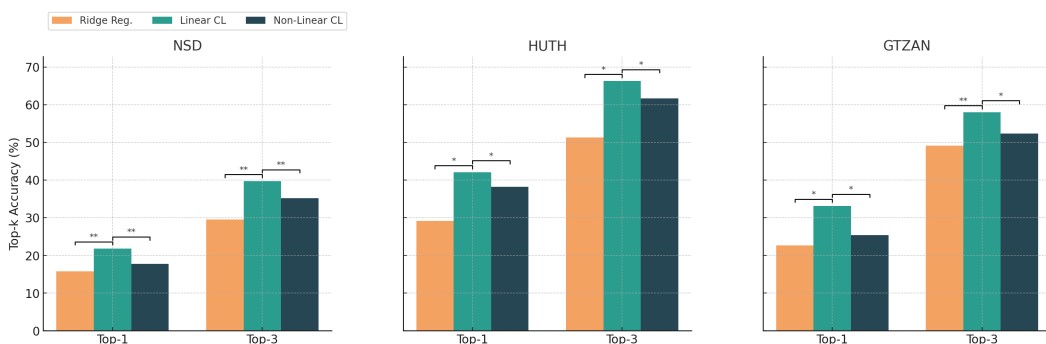

Figure 3: Quantitative bar charts to visualize decoding results. Stars above the bars reveal significance, according to the table in the Appendix. Double stars indicate pvalue lower than 1e-10.

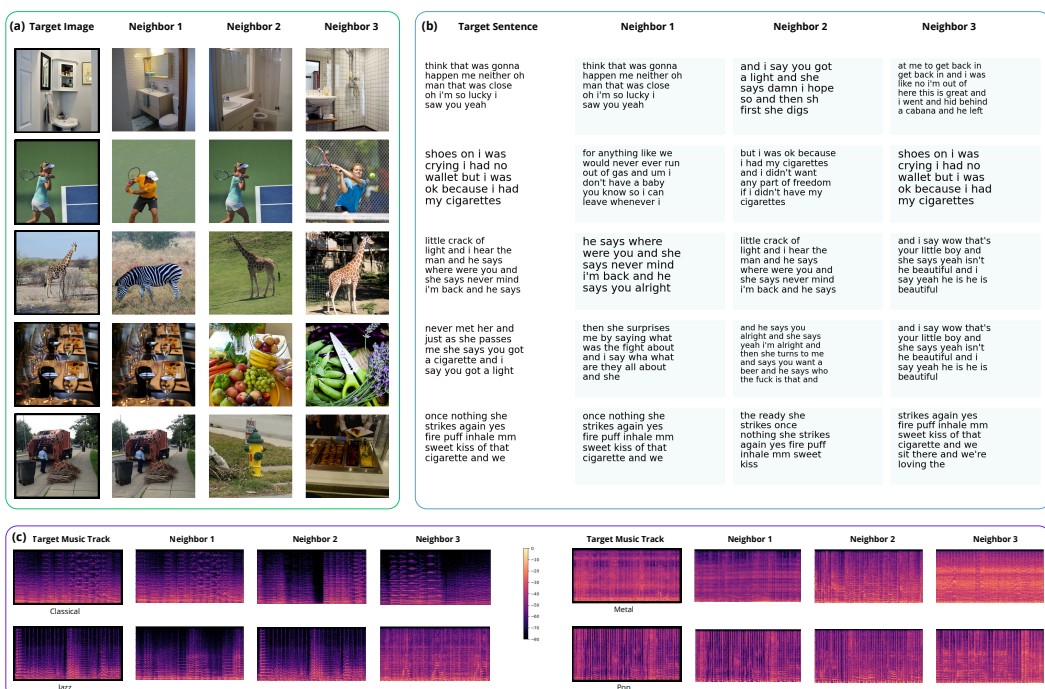

Figure 4: Random samples of brain decoding results. For each panel, the target column shows the ground-truth stimulus (music track, image, or sentence, depending on the modality), while the neighbor columns display the top retrieved candidates from the model's latent space based on cosine similarity. (a) Retrieval of images viewed by participants. (b) Retrieval of text/sentences corresponding to the neural response. (c) Retrieval of music tracks from brain activity. These qualitative examples illustrate that the predicted neural embeddings often retrieve semantically related stimuli, highlighting the model's ability to capture meaningful structure in brain representations.

In Figure 4, we present qualitative decoding results from the test sets of the three datasets. In all cases, the retrieved samples show clear conceptual similarity with the target stimulus. For the NSD dataset, the retrieved images capture semantic content consistent with the reference, such as animals, food, or sports. A similar trend is observed in the textual modality, where the retrieved sentences convey the same high-level meaning as the ground-truth sentences. Finally, for the music dataset, the comparison of spectrograms highlights modality-specific correspondences: for instance, in the jazz genre case, the retrieved samples share distinctive frequency patterns visible in the target

spectrogram; in contrast, for the metal sample, the retrieved spectrograms display higher energy across frequencies, reflecting the different acoustic structure of the genre.

## 5 DISCUSSION

Our results indicate that a linear mapping trained with a contrastive objective is a robust and general strategy for fMRI decoding across modalities. Below we discuss two central findings — the superiority of contrastive learning over ridge regression, and the consistent advantage of linear over shallow non-linear mappings — and offer mechanistic explanations consistent with prior literature.

### 5.1 CONTRASTIVE LEARNING FOR VECTORIAL REPRESENTATION OF CONCEPTS

If concepts in the brain are organized as vectors in high-dimensional spaces (Piantadosi & et al., 2024), then meaning is carried primarily by their relative geometry: distances, angles, and directions capture similarity, typicality, and relational structure. This view suggests that learning and comparison are fundamentally geometric operations (Ferrante et al., 2025). Contrastive objectives (Chen et al., 2020) directly operationalize this principle by maximizing angular similarity between matched pairs while enforcing separation from distractors, effectively aligning the training loss with the retrieval metric. In practice, the negative set acts as a data-driven regularizer: it suppresses directions that reflect nuisance variance in fMRI but are not discriminative in the target space, while amplifying those aligned with semantic information. Our findings that contrastive mappings systematically outperform ridge regression support this interpretation. Whereas ridge minimizes point-wise $\ell_2$ error—implicitly prioritizing magnitude alignment—contrastive learning preserves and sharpens the relational structure of the embedding manifold. This can be seen as a form of "reverse engineering" of the brain's representational optimization, consistent with contemporary accounts of concepts as vectors whose relations, rather than absolute values, encode meaning.

Notably, the same advantage holds when mapping into embedding spaces that are not trained contrastively, such as *LLaMA3* language embeddings. This indicates that the benefit is not merely 'contrastive-to-contrastive alignment', but a more general advantage of geometry-aware objectives in aligning fMRI to vector spaces.

### 5.2 LINEAR VS NON-LINEAR MAPPING

A second central finding is the consistent superiority of linear over shallow non-linear mappings. While this may appear counterintuitive, given the assumption that non-linear networks are needed to capture neural complexity, large-scale analyses of fMRI show that linear models perform better at the macroscale (Nozari et al., 2024; Schulz et al., 2020). This apparent linearity has a principled explanation: although neurons are individually non-linear, fMRI signals reflect averages over millions of units, filtered in time and further smoothed by observation noise. These operations suppress higher-order dynamics and yield an effective signal that approximates a first-order (Taylor-linear) expansion of the underlying neural processes. In this setting, linear mappings are not just a simplifying choice but may be the most explainable and faithful representation of the observable data. Our experiments are consistent with this view: non-linear layers, while increasing expressivity, also relax the inductive bias that preserves embedding geometry. In the small-to-moderate data regime typical of fMRI, this flexibility can rotate or distort informative directions, amplify noise-driven variance, and disrupt calibration of vector norms, ultimately degrading retrieval accuracy. Of course, the space of possible non-linear architectures is vast, and we cannot exclude that specific designs or training regimes may close this gap.

Taken together, these perspectives highlight a broader principle: when decoding with rich, pretrained representations, most of the relevant non-linear structure is already embedded upstream in the foundation models. The decoder's role is not to discover new features but to align noisy brain measurements with an existing embedding geometry. Linear contrastive mappings are therefore well suited: they provide stable optimization, suppress nuisance variance, and maximize discriminative alignment. Conceptually, they instantiate two converging ideas — that semantic information in the brain is organized in vector spaces, and that fMRI provides a linearized view of these representations.

### 5.3 LIMITATIONS

Several limitations of this work should be acknowledged. First, while our analyses systematically compared linear, ridge, and shallow non-linear decoders, the space of non-linear architectures is large. It remains possible that alternative designs or more extensive hyperparameter sweeps could get better performance. However, this consideration also reinforces our main claim: in practice, the computational cost of exhaustively searching for an optimal non-linear configuration may not be justified, since strong performance can already be achieved with simple linear contrastive approaches. Additionally, we did not evaluate time-aware architectures (e.g., LSTMs, Transformers) since temporal structure has been removed by design (GLM betas and HRF-averaged): exploring these directions—particularly in minimally averaged fMRI—is an important avenue for future work. Second, our study does not directly address generalization across datasets. Decoding models were trained and evaluated within individual modalities, leaving open the question of how well such mappings transfer across datasets. Finally, decoding brain activity into rich semantic spaces raises concerns about potential misuse, especially if applied to unconstrained settings or without informed consent. Future work should be guided not only by scientific objectives but also by principled discussions of data governance, individual rights, and ethical safeguards (Yuste et al., 2017).

## 6 CONCLUSIONS

We presented a unified framework for fMRI decoding that maps neural responses into the embedding spaces of large foundation models and we evaluated it across three distinct modalities (vision, language, and music) using the same pipeline. Empirically, a very simple strategy—a subject-aligned linear mapping trained with a standard contrastive objective (NT-Xent)—consistently outperforms both ridge regression and non-linear MLP decoders, yielding clear improvements of roughly 10–15% in Top-1 and Top-3 retrieval accuracy across datasets. The core contribution of this work therefore lies less in algorithmic novelty and more in the strength of the evidence and the methodological message it supports: *"do not overcomplicate—contrastive alignment with a linear decoder often suffices"*. While our study focuses on retrieval-based decoding, the results provide a strong, motivated and reproducible approach for future work that may extend contrastive-linear alignment to generative reconstruction, cross-modality transfer, and integration with higher-resolution neural measurements.

Our central claim is not that linear models are universally optimal, but that—under standard fMRI preprocessing and matched data budgets—the choice of loss function (contrastive vs MSE) has a far greater impact on decoding performance than architectural depth.

### ETHICS STATEMENT

This study makes exclusive use of publicly available fMRI datasets (Allen et al., 2022; LeBel et al., 2023; Nakai et al., 2022), which were collected with informed consent under protocols approved by the respective institutional review boards. No new human data were collected. Potential risks of brain decoding research, such as privacy concerns and possible misuse, are acknowledged. The work is intended solely to advance scientific understanding and should not be used for individual-level prediction or surveillance.

### REPRODUCIBILITY STATEMENT

Preprocessing pipelines, model architecture, training objectives, and hyperparameters are detailed in the manuscript (Section 3 & 4). All experiments can be reproduced with the scripts provided as a zipped repository in Supplementary Materials. We highlight that large language models (LLMs) were used exclusively for textual editing and polish writing.

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

# A APPENDIX

### A1. STATISTICAL SIGNIFICANCE AND EFFECT SIZES

We quantified the statistical significance of the improvements reported in Table 1 of the main paper. For each dataset and metric, we performed paired $t$-tests across subjects and random seeds, comparing the Linear-CL decoder against both Ridge regression and the non-linear MLP. Effect sizes are reported as paired Cohen's $d$. See table A1.

Table A1: $t$-tests (Linear-CL vs. Ridge, Linear-CL vs. Non-Linear MLP) on Top-1 and Top-3.

| Dataset | Metric | Comparison | $t$ | $p$ | $d$ |
|---------|--------|------------|-----|-----|-----|
| NSD (Image) | Top-1 | Linear vs Ridge | 14.25 | $1.3 \times 10^{-11}$ | 2.59 |
| | Top-3 | Linear vs Ridge | 17.78 | $2.6 \times 10^{-13}$ | 2.98 |
| | Top-1 | Linear vs Non-Linear | 19.03 | $7.8 \times 10^{-14}$ | 3.26 |
| | Top-3 | Linear vs Non-Linear | 17.97 | $2.2 \times 10^{-13}$ | 3.02 |
| HUTH (Language) | Top-1 | Linear vs Ridge | 5.110 | $4.4 \times 10^{-4}$ | 1.23 |
| | Top-3 | Linear vs Ridge | 6.780 | $2.6 \times 10^{-5}$ | 1.40 |
| | Top-1 | Linear vs Non-Linear | 2.210 | $4.4 \times 10^{-2}$ | 0.59 |
| | Top-3 | Linear vs Non-Linear | 3.702 | $2.3 \times 10^{-3}$ | 0.92 |
| GTZAN (Music) | Top-1 | Linear vs Ridge | 7.740 | $5.2 \times 10^{-8}$ | 1.55 |
| | Top-3 | Linear vs Ridge | 12.96 | $5.1 \times 10^{-13}$ | 2.79 |
| | Top-1 | Linear vs Non-Linear | 8.358 | $1.4 \times 10^{-8}$ | 1.67 |
| | Top-3 | Linear vs Non-Linear | 6.425 | $1.2 \times 10^{-6}$ | 1.28 |

### A2. RECONSTRUCTION ERROR (MSE) ACROSS MODELS

We also report (in Table A2) the Mean Squared Error (MSE) between predicted and ground-truth embeddings for all datasets. As expected, Ridge regression (explicitly optimized for MSE) achieves the lowest reconstruction error, while contrastive models exhibit higher MSE despite superior retrieval performance.

Table A2: Mean Squared Error (MSE; mean $\pm$ std. across subjects) for Ridge, Non-Linear MLP, and Linear-CL decoders.

| Dataset | Model | MSE (mean $\pm$ std.) |
|---|---|---|
| HUTH (Language) | Ridge | $1.02 \pm 0.02$ |
| | Non-Linear | $8.28 \pm 0.97$ |
| | Linear-CL | $33.17 \pm 1.62$ |
| NSD (Image) | Ridge | $0.188 \pm 0.003$ |
| | Non-Linear | $0.216 \pm 0.002$ |
| | Linear-CL | $0.262 \pm 0.003$ |
| GTZAN (Music) | Ridge | $0.00141 \pm 0.00013$ |
| | Non-Linear | $0.219 \pm 0.020$ |
| | Linear-CL | $0.779 \pm 0.090$ |

Table A3: Ablation of the subject-alignment layer $A_k$. "Anatomical-CL" denotes a model trained separately per subject without the alignment layer; Linear-CL is the full multi-subject model with learned $A_k$.

| Dataset | Metric | Linear-CL (%) | Anatomical-CL (%) |
|---|---|---|---|
| NSD (Image) | Top-1 | $21.80 \pm 0.76$ | $17.8 \pm 1.11$ |
| | Top-3 | $39.66 \pm 0.91$ | $34.5 \pm 1.49$ |
| HUTH (Language) | Top-1 | $42.04 \pm 2.19$ | $37.2 \pm 2.55$ |
| | Top-3 | $66.25 \pm 2.87$ | $61.9 \pm 2.81$ |
| GTZAN (Music) | Top-1 | $33.13 \pm 1.47$ | $27.4 \pm 1.69$ |
| | Top-3 | $57.97 \pm 1.12$ | $50.5 \pm 1.92$ |

These results highlight that lower MSE does not necessarily translate into better retrieval, since the contrastive objective is invariant under global rescaling of embeddings and optimizes relative similarity rather than absolute reconstruction error.

## A3. ABLATION OF THE SUBJECT-ALIGNMENT LAYER

Generally, fMRI responses exhibit strong inter-subject variability in both anatomical organization and functional topography. As shown in recent cross-subject decoding frameworks (Tang & Huth, 2025; Ferrante et al., 2024a; d'Ascoli et al., 2025; Aggarwal et al., 2024; Thual et al., 2023), anatomical alignment alone is insufficient for high-level semantic decoding; a subject-specific functional alignment is typically required to map different brains into a shared representational space. Our alignment layer $A_k$ plays exactly this role. Rather than being applied as a separate preprocessing step, it is jointly optimized inside the contrastive objective, allowing the model to learn linear subject-specific transformations (matrix) that project each subject's fMRI activity into a common functional space in which stimulus representations are comparable.

We evaluated the contribution of the subject-specific alignment matrices $A_k$. Removing the alignment layer corresponds to training a separate anatomical-only model per subject. This ablation reduces performance across all datasets (Table A3), confirming the benefit of learning a shared functional space.

## A4. IMAGE-LEVEL SIMILARITY METRICS

To facilitate comparison with recent generative non-linear decoders for vision stimuli (Mind-Eye2/UMBRAE from Xia et al. (2024b) and Scotti et al. (2024)), we computed low- and high-level similarity metrics between the ground-truth NSD test images and the images retrieved by our Linear-CL model. Although our method is retrieval-based rather than generative, the scores are competitive relative to the ranges reported in MindEye2/UMBRAE.

Table A4: Similarity metrics between ground-truth NSD test images and images retrieved by the Linear-CL decoder (mean $\pm$ std. across test samples).

| Metric | Value (mean $\pm$ std.) |
|---|---|
| AlexNet (layer 2) | $0.9408 \pm 0.0649$ |
| AlexNet (layer 5) | $0.8926 \pm 0.0869$ |
| CLIP similarity | $0.9163 \pm 0.1244$ |
| SSIM | $0.4993 \pm 0.3340$ |
| Pixel-wise correlation (PixCorr) | $0.3540 \pm 0.2369$ |
| EfficientNet-B1 distance | $0.6682 \pm 0.2397$ |

A5. HYPERPARAMETER EXPLORATION

For completeness, we report the additional hyperparameter analyses (Table A5). We use the standard NT-Xent sampling scheme, with one positive pair per anchor and all other items in the batch acting as negatives, as in SimCLR and CLIP. The temperature parameter $\tau$ was tuned separately for each dataset. The values reported correspond to the best temperature per dataset and were stable across random seeds. We also tested multiple batch sizes, observing that performance consistently improved for larger batches, consistent with the behavior of contrastive losses where a larger number of in-batch negatives improves the estimation of the objective and stabilizes optimization.

The best model with "Identity" as activation function is not intended to represent a separate non-linear architecture. It is in fact the same architecture used in our Linear-CL model: a stack of linear layers with Identity activations, which collapses to a linear transformation. This is precisely our model that achieves the best performance. Importantly, this "multi-layer linear network" corresponds to a low-rank factorization of the full linear weight matrix. In practice, this acts as an additional form of regularization: the transformation is still linear, but decomposed into smaller matrices.

Table A5: Hyperparameter search space (top) and best values per dataset (bottom).

| Hyperparameter | Values explored |
|---|---|
| Hidden dim. | {4096, 2048, 1024} |
| Activation func. | {Identity, ReLU, GELU} |
| Num. Layers | {1, 2, 5} |
| Learning Rate | {1e-3, 1e-4, 1e-5} |
| Temperature $\tau$ | {0.02, 0.05, 0.10, 0.50} |
| Weight Decay | {1e-3, 1e-4} |
| Batch Size | {128, 512, 1024, 2048} |

| Dataset | Activation | Hidden | Layers | LR | $\tau$ | WD | BS |
|---|---|---|---|---|---|---|---|
| NSD (Image) | Identity | 1024 | 2 | 1e-4 | 0.02 | 1e-3 | 2048 |
| HUTH (Lang.) | Identity | 2048 | 2 | 1e-3 | 0.05 | 1e-4 | 1024 |
| GTZAN (Music) | Identity | 1024 | 1 | 1e-3 | 0.10 | 1e-4 | 2048 |

