# OpenReview forum: "Linear Maps, Contrastive Objectives: A Principled Strategy for fMRI Decoding Consistent Across Modalities"
_ICLR.cc/2026/Conference — ICLR 2026 Conference Desk Rejected Submission_

### Official Review · Reviewer_Rfki · 2025-10-29

**Soundness:** 1
**Presentation:** 2
**Contribution:** 2
**Rating:** 4
**Confidence:** 4

**Summary:**

The paper argues that when decoding fMRI into the representation spaces of large foundation models, a simple linear mapping trained with a contrastive objective is consistently better than ridge regression and shallow MLPs. The authors evaluate three public datasets, and use retrieval in embedding space as the metric. A small architecture with subject-specific linear alignment followed by a linear projection yields Top-1 gains over ridge and also outperforms their non-linear MLP across all modalities.

**Strengths:**

The paper’s key strength is its evidence that a simple linear mapping is adequate for aligning fMRI signals with semantically rich embedding spaces (e.g., from multimodal foundation models). This is a surprising and consequential observation, suggesting that the geometry of brain-recorded activity admits an approximately linear relationship to these representations, thereby refining our understanding of both fMRI signal structure and neural encoding at the population level.

**Weaknesses:**

Although the conclusion—that an efficient linear mapping suffices for fMRI decoding—is intriguing, the experimental design supporting this claim appears insufficiently rigorous:

1. The paper compares ridge regression (trained with an MSE objective and L2 regularization ?) to the proposed linear model trained with a contrastive (InfoNCE-style) objective. Because the target embedding (e.g., CLIP) itself is learned with a contrastive loss, this comparison is not fair: one would expect a method trained with a geometry-aligned contrastive loss to outperform MSE by construction.

2. The paper states that an MLP underperforms the linear model (Table 1), yet Table 2 shows that **Identity** activation yields the best MLP performance. Theoretically, an MLP with identity activations collapses to a linear map; with sufficient training and comparable regularization, it should match (or closely approach) the linear model. So does the comparsion of MLP with identity activation and linear model make sense?

3. The proposed linear model includes a subject-specific adjustment matrix (A_k). It is unclear whether the MLP baseline incorporates an analogous adjustment . The paper should document these details (main text or appendix).

I am willing to increase my score if the authors address the issues above.

**Questions:**

1. Please according to the soundness issues in Weaknesses part
2. It is recommend that the author could explain more why linear model is better than other more complex methods.

---

> ### Author Response · Authors · 2025-11-21
>
> We thank the reviewer for their insightful and constructive feedback. Below we address each point.
>
> **Regarding Weakness 1**: Thanks for the comment. We agree that a contrastive objective is naturally more aligned with the geometry of CLIP embeddings. However, this does not invalidate the comparison. CLIP-like multimodal embeddings are the standard target space in contemporary brain decoding, and ridge regression with an MSE objective is exactly the baseline used in prior work for mapping fMRI into these spaces. The question we aim to address is whether a geometry-aware loss improves decoding beyond the commonly used MSE-based linear mapping.
> Importantly, the advantage of the contrastive formulation is not specific to CLIP. We validated this using alternative target embeddings that are not trained with contrastive objectives—for example, LLAMA-derived language embeddings. These representations are trained autoregressively, yet they still exhibit cosine-structured semantic geometry due to the attention mechanism. In this setting, the contrastive objective again outperforms MSE, indicating that the improvement is not simply “contrastive-to-contrastive alignment,” but that contrastive learning provides a better inductive bias for aligning brain activity with any embedding space that encodes relational similarity.
> As reported in Table 1 of the paper, Linear CL for LLAMA3 prediction improves retrieval substantially over ridge regression:
> Top-1: Linear CL = 42.04%, Ridge = 29.11%;
> Top-3: Linear CL = 66.25%, Ridge = 51.33%
>
> This is also conceptually intuitive: by training with a contrastive loss, the brain activity becomes treated as just another modality that should inhabit the same semantic space as text or images. The objective encourages fMRI representations to adopt the same geometric structure as other modalities—precisely the type of relational organization hypothesized in cognitive models of vector-space representations (e.g., Piantadosi et al. 2024).
>
> **Regarding Weakness 2**: Thanks for the comment. We realize that our terminology may have caused confusion. The model labeled as “MLP + Identity activation” in Table 2 (now Table A.5 in Appendix) is not intended to represent a separate non-linear architecture. It is in fact the same architecture used in our Linear-CL model: a stack of linear layers with Identity activations, which collapses to a linear transformation. This is precisely our model (described in Section 3.4) that achieves the best performance.
> By contrast, the models referred to as “non-linear CL” throughout the paper are the MLPs with ReLU activations. These introduce genuine non-linear transformations and are the ones that systematically underperform the linear version across datasets.
> Importantly, our best model corresponds to a low-rank factorization of the full linear weight matrix. In practice, this acts as an additional form of regularization: the transformation is still linear, but decomposed into smaller matrices. See Appendix A.5.
>
> **Regarding Weakness 3**: Thanks for the comment. We apologize if the description was unclear. Non-linear approaches do incorporate the same subject-specific adjustment matrix Ak. This ensures that the comparison is evaluated under identical cross-subject conditions. Thus, the subject-alignment component is not unique to the proposed model—it is shared with non-linear methods. This was done intentionally to isolate the effect of architectural choices rather than confound them with alignment differences.
>
> **Regarding Question 2**: Thanks for the suggestion. Our result is consistent with recent theoretical and empirical findings in systems neuroscience—particularly the work of Nozari et al. (2024), which shows that at the spatial and temporal scale of fMRI, neural population dynamics become effectively linearized. In other words, the BOLD signal behaves as a first-order approximation of the underlying neural processes due to spatial averaging, hemodynamic filtering, and noise. This observation is also fully aligned with foundational work in the field. Classic frameworks for encoding and decoding fMRI signals—most notably Naselaris et al. (2011)—already emphasized that linear mappings capture the dominant structure linking stimuli and voxel-wise responses. Although the field has since advanced, this principle remains central: the transformation from stimulus features to BOLD responses behaves approximately linearly at the macroscopic scale. Under this regime, we empirically found that:
> ●	Linear mappings are sufficient to capture the dominant stimulus–response relation.
> ●	Additional non-linear capacity tends to amplify noise, especially in high-dimensional, low-sample neuroimaging settings.
> ●	This effect is even more pronounced in contrastive learning: the model must preserve relative similarity, and linear mappings retain this structure more reliably than deeper networks.

---

### Official Review · Reviewer_wpk1 · 2025-10-30

**Soundness:** 2
**Presentation:** 2
**Contribution:** 2
**Rating:** 2
**Confidence:** 4

**Summary:**

This paper investigates fMRI decoding by mapping brain activity to the embedding spaces of foundation models for three different modalities: vision, language, and audio. The authors systematically compare three types of decoders: ridge regression, a linear mapping trained with a contrastive objective, and a shallow MLP with non-linearities also trained with a contrastive objective. Their experiments across three public datasets show that the linear contrastive model consistently achieves the highest retrieval accuracy, outperforming both ridge regression and the non-linear model. The authors argue that this supports two main ideas: 1) the contrastive objective is more suitable for this task than the point-wise error minimization of ridge regression, and 2) linear models are sufficient and often superior for decoding macroscopic fMRI signals, whose inherent noise and averaging properties effectively linearize the underlying neural computations.

**Strengths:**

- The paper's main strength is its consistent application and evaluation of the same decoding pipeline across three distinct and important modalities (vision, language, music). This provides strong evidence for the generalizability of their findings.

- The work delivers a clear and practical message: a simple, subject-aligned linear model trained with a contrastive loss is a robust and effective strategy for fMRI decoding. This provides a strong baseline for future research.

- The authors ground their approach in plausible theories from cognitive science (vector representations of concepts) and neuroscience (the linear nature of macroscopic fMRI signals), providing a solid theoretical motivation for their empirical results.

**Weaknesses:**

The paper's conclusions, while interesting, are weakened by several methodological omissions, overstated claims, and presentation issues.

- The central claim that the linear contrastive (Linear CL) model "outperforms" ridge regression is based solely on retrieval accuracy. This is unsurprising, as the Linear CL model is directly optimized for a retrieval-based contrastive loss, while ridge regression minimizes Mean Squared Error (MSE). A fairer comparison would require evaluating both models on both types of metrics (e.g., reporting MSE for the CL models and showing feature-wise correlation for Ridge in Fig 3b). Without this, the claim of superiority is not fully substantiated.

- The paper makes a strong claim that "architectural complexity does not translate into performance gains." However, this conclusion is based on comparing only to a shallow MLP. The space of non-linear architectures is vast, and other work (e.g., Careil et al., 2025, "Dynadiff", Table 3) has shown significant gains with more complex non-linear models for the same task. The current experiments are insufficient to support such a general conclusion.

- Critical methodological details are missing:
    - The paper never specifies the size of the retrieval set (the reference set Rs) for the Top-k accuracy evaluation. Retrieval accuracy is meaningless without knowing the number of distractors (e.g., Top-1 accuracy of 42% is very different if the pool size is 10 vs. 1000). This is a critical omission that prevents proper interpretation of the main results in Table 1.
    - The use of a subject-specific alignment layer is mentioned, but its purpose is unclear, especially since the evaluation is then described as being performed "within subject." The details of the inter-subject alignment (e.g., anatomical or functional) and its role in the model are not sufficiently explained.
    - The paper does not justify why different numbers of voxels were selected for each dataset (15.7k for images, 10k for language, 3k for music), making it difficult to assess the consistency of the approach.

Minor comments
- The paper is missing citations to highly relevant recent work in fMRI decoding (e.g. Careil et al., 2025; Aggarwal et al., 2024; Thual et al., 2023; Banville et al., 2025; Ye et al., 2025; Tang et al, 2023; Jalouzot et al., 2025).
- The results in Table 1 are difficult to parse; a bar chart would be much clearer for comparing the three methods across different conditions.

**Questions:**

- Could you clarify the multi-subject alignment procedure and its purpose?

- To provide a fairer comparison with ridge regression, could you report the Mean Squared Error (MSE) for all tested models? This would help disentangle performance gains due to the objective function versus the architectural choice.

- How are the evaluation retrieval sets built and what are their size for the three datasets? This information is essential for interpreting the reported accuracy percentages.

---

> ### Author Response · Authors · 2025-11-21
>
> We thank the reviewer for their insightful and constructive feedback. Below we address each point.
>
> **Regarding Weakness 1**: We computed the Mean Squared Error (MSE) for all models. As expected, Ridge—which optimizes MSE directly—achieves the lowest error, while contrastive models show higher MSE. This is not contradictory: contrastive learning is invariant to global rescaling of embeddings, meaning that MSE can vary substantially even when retrieval performance is unchanged. Since our task is stimulus identification from fMRI, the relevant metric is relative similarity, not absolute reconstruction. Ridge obtains low MSE but substantially worse retrieval accuracy, confirming the mismatch between objective and evaluation. Full MSE results are reported in Appendix A.2.
>
> **Regarding Weakness 2**:
> Across all datasets, adding non-linearities, depth, or residual connections consistently reduces accuracy relative to the linear contrastive decoder. This aligns with known behavior in low-sample, high-dimensional neuroimaging regression, where extra capacity tends to amplify noise. This does not imply that no non-linear model could ever do better; rather, simple increases in depth or capacity are not sufficient in this regime.
> Architectures such as Dynadiff (Careil et al., 2025) address a different task, involving (i) a deep diffusion decoder predicting latent trajectories, (ii) cross-attention between neural responses and visual tokens, (iii) large-scale image pretraining, and (iv) a generative reconstruction objective. A direct comparison with models such as Dynadiff is not straightforward but, for completeness, we computed the same similarity scores between ground-truth images and the images retrieved by our model. Despite the conceptual mismatch, our retrieval-based decoder achieves values in a similar range for NSD (see Appendix A.4).
>
> **Regarding Weakness 3, Point 1**: We apologize for the omission, the size of the retrieval set has now been clearly specified in the revised manuscript (Section 3):
>
> • NSD: ~980 images in the test → chance Top-1 ≈0.10%, Top-3 ≈0.31%.
> • HUTH: 250 held-out sentences → chance Top-1 =0.40%, Top-3 =1.20%.
> • GTZAN: 60 musical excerpts → chance Top-1 =1.67%, Top-3 =5.00%.
>
> **Regarding Weakness 3, Point 2**: We clarify the role of the subject-specific alignment layer Ak. fMRI responses exhibit strong inter-subject variability in both anatomical organization and functional topography. As shown in recent cross-subject decoding frameworks such as TRIBE (d’Ascoli et al. 2025) and Tang et al. 2025, anatomical alignment alone is insufficient for high-level semantic decoding; a subject-specific functional alignment is typically required to map different brains into a shared representational space. Our alignment layer Ak plays exactly this role. Rather than being applied as a separate preprocessing step, it is jointly optimized inside the contrastive objective, allowing the model to learn linear subject-specific transformations that project each subject’s fMRI activity into a common functional space in which stimulus representations are comparable. This is similar in spirit to linear functional alignment methods (“Through Their Eyes” from Ferrante et al. 2024), but integrated end-to-end with the contrastive learning framework. Although evaluation is performed “within subject”, training is performed across all subjects simultaneously. Removing 𝐴k​ corresponds to training separate models per subject, and results are provided in Appendix A.3.
>
> **Regarding Weakness 3, Point 3**: The voxel selection follows the procedure standardly used in the corresponding literature (added in Section 3). For the NSD we follow the protocol introduced in Allen et al. (Nature Neuroscience, 2022), which provides subject-specific maps. The recommended practice—used in all major NSD decoding papers (e.g., Takagi & Nishimoto 2023)—is to retain the most reliable voxels in V1–V4, posterior parietal cortex, and high-level ventral stream.
> For the narrative language dataset (HUTH), we follow prior work on voxel filtering in language decoding (e.g., Tang et al. 2023; Toneva & Wehbe 2019). We select voxels with an encoding model mapping from stimulus embeddings to fMRI responses and keep voxels with higher predictive correlation on held-out data.
> For music decoding (GTZAN), we use voxelwise encoding accuracy to isolate auditory- and music-responsive regions. We trained a voxelwise encoder (similar approach for HUTH dataset) using CLAP-audio features and applied a correlation threshold. This resulted in ∼3k voxels localized in STG, HG, IFG, and premotor cortex. See “R&B-rhythm and brain” paper from Ferrante et al. 2024 for details.
>
> **Regarding Minor Comment 1**: Thanks for the comment. We added the suggested references to strengthen contextualization.
>
> **Regarding Minor Comment 2**: We added a bar chart of decoding results in new Figure 3.

---

> > ### Comment · Reviewer_wpk1 · 2025-11-26
> >
> > I would like to thank the authors, I appreciate their effort to address my different concerns.  I have remaining issues which I detail below.
> >
> > **Weakness 1**
> >
> > > As expected, Ridge—which optimizes MSE directly—achieves the lowest error, while contrastive models show higher MSE.
> > > ...
> > > Since our task is stimulus identification from fMRI, the relevant metric is relative similarity, not absolute reconstruction.
> >
> > This is indeed the main approach to fMRI decoding today to address training without over-fitting in those *low* data regimes.
> > Ridge is an interesting and informative baseline for such an objective but it should not be considered a strong baseline nor SOTA as it does not optimize for this task. I still think that claiming that contrastive decoders *outperform* Ridge regression is not valid and that the text should be revised.
> >
> > **Weakness 2**
> >
> > > Architectures such as Dynadiff (Careil et al., 2025) address a different task ...
> >
> > Indeed, MindEye2 was the correct example here and not Dynadiff.  I would like to apologize for this mistake and the resulting confusion.  In the MindEye2 paper (Scotti et al. 2024), they report \~98% retrieval accuracy (Table 1). Since they use non-linear decoders with contrastive learning, this is in contradiction with the claims in the paper. Could the authors try to reconcile both results?
> >
> > **Weakness 3.1**
> >
> > > • NSD: \~980 images in the test → chance Top-1 ≈0.10%, Top-3 ≈0.31%. • HUTH: 250 held-out sentences → chance Top-1 =0.40%, Top-3 =1.20%. • GTZAN: 60 musical excerpts → chance Top-1 =1.67%, Top-3 =5.00%.
> >
> > Thanks for reporting those values. They are very different across datasets which prevents comparisons between them.  Even though the latter is not critical for the paper, could the authors explain this choice?
> >
> > **Weakness 3.2**
> >
> > The problem with the multi-subject setup is that it complicates the framework while not being necessary to support the claims of the paper.  For instance, as mentioned by  the decoders have multiple layers even without non-linearities.  One resulting problem I see is that the hidden dimension *h* is smaller than the target embeddings dimension (4096 if using LLama 3 8B, to be confirmed, c.f. comment below) only for the language dataset.  This means that the decoder is actually reduced-rank/bottlenecked while it is not for the others.
> > I believe the authors should have trained and evaluated one subject at a time without alignment layers, with a consistent one-layer model or a consistent bottleneck model.
> >
> > **Comment**
> >
> > The language target embeddings construction is not described in the paper. Section 3.2 describes a way of constructing word embeddings which are then used for voxel selection.  Are those the same embeddings used for training and retrieval?
> >
> > **Final remark**
> >
> > I appreciate the authors' effort to address the concerns of the different reviewers.  The remaining issues are too significant, I will only slightly increase my score. I still think the paper is not ready for acceptance in its current form.

---

> ### Author Response · Authors · 2025-11-26
>
> **Weakness 1**
>
> We thank the reviewer for this clarification. We agree that Ridge is not a strong baseline for identification, since it optimizes MSE rather than discriminative similarity. Our intention was not to claim that Ridge is SOTA for this task, but to show empirically that (even under matched training conditions) contrastive decoders achieve substantially better identification performance than Ridge, which is exactly the metric relevant for fMRI decoding. This is not presented as evidence that Ridge is “strong” or “SOTA” (even if it remains the most widely used approach in brain decoding), but simply to highlight that minimizing MSE does not translate into better identification, further supporting the need for contrastive objectives in low-sample, high-dimensional fMRI decoding.
>
> **Weakness 2**
>
> While MindEye2 indeed reports very high retrieval accuracy, these results stem from architectural and experimental choices that substantially influence the decoding problem and therefore do not contradict our findings. Although both our work and MindEye2 use cross-subject information, the nature of this approach is fundamentally different.
> - A highly structured and linearizable target space: MindEye2 maps to OpenCLIP ViT-bigG/14, a very smooth and high-dimensional embedding space, further regularized by a diffusion prior. This simplifies retrieval compared to our direct decoding into CLIP embeddings without such priors.
> - Reduced retrieval pool: Retrieval is evaluated over 300 candidates (Table 1) rather than the full NSD test set (~1000 images), reducing task difficulty by ~3×.
> - Dedicated retrieval head: MindEye2 employs a separate contrastive submodule trained with BiMixCo + SoftCLIP, operating on embeddings already shaped by the diffusion prior and MLP backbone, very different from assessing the intrinsic quality of the fMRI→embedding mapping, which is the focus of our work.
>
> For these reasons, MindEye2’s high retrieval accuracy reflects large-scale pretraining, heavy averaging, and a highly structured target space—not an inherent advantage of non-linear decoders in low-sample fMRI decoding.
>
> **Weakness 3.1**
>
> The differences in chance levels arise from the intrinsic structure of each dataset, and our goal was to evaluate performance within each modality rather than compare modalities directly. Specifically, the size of the test set is fixed by the original experimental design of each dataset (NSD: ~980 images; HUTH: 250 sentences; GTZAN: 60 audio excerpts), and we follow the standard protocol used in prior work for each domain.
>
> **Weakness 3.2**
>
> Multi-subject alignment: As described in Sec. 3.4, we use only a subject-specific linear matrix, followed by a shared linear decoder. This is not an added complexity: it is a single linear map factored into matrices. Appendix A3 shows that removing this alignment (“Anatomical-CL”) consistently reduces performance across all datasets (Table A3), confirming that this step is beneficial yet remains strictly linear.
> Linear decoder and bottleneck: The decoder has multiple linear layers but, as noted in Appendix A5, without non-linearities it remains mathematically equivalent to a single linear projection (a low-rank factorization). The hidden dimension h is shared across subjects for consistency, but not across datasets because they are esaminated independently (cross-domain is not our claim). Since LLaMA3 embeddings are 4096-dimensional, this creates a bottleneck only in HUTH dataset, but it affects both the linear and non-linear decoders identically (Appendix A5), ensuring a fair comparison within each dataset.
>
> **Comment**
>
> Finally, the LLaMA3 embeddings described in Sec. 3.2 are exactly the same embeddings used for decoder training and retrieval; they are not only used for voxel selection. Sorry for not being clear in this description.
>
> **Final Remark**
>
> We thank the reviewer for the careful assessment and for acknowledging our efforts to address the concerns raised. We will revise the manuscript accordingly to improve clarity, streamline the exposition, and emphasize the scope and limitations of our contributions. We appreciate the reviewer’s feedback and believe that the revisions will substantially strengthen the final version of the paper.

---

### Official Review · Reviewer_PmyW · 2025-10-31

**Soundness:** 3
**Presentation:** 3
**Contribution:** 3
**Rating:** 6
**Confidence:** 3

**Summary:**

This paper investigates the role of linear versus non-linear mappings for decoding fMRI activity into the embedding spaces of foundation models (CLIP, CLAP, LLaMA). Across three modalities—vision, language, and music—the authors demonstrate that a simple linear contrastive decoder consistently outperforms both ridge regression and non-linear MLP decoders. The study argues that due to spatial and temporal averaging, fMRI signals behave effectively linearly, making contrastive-linear objectives both biologically plausible and empirically optimal for brain decoding.

**Strengths:**

1. The paper builds a principled connection between cognitive theories of vector-based concept representation and the observed linearity of fMRI signals, offering a coherent rationale for why linear contrastive decoders should work.

2. The writing is clear, figures are informative, and the narrative effectively bridges neuroscience and machine learning perspectives.

**Weaknesses:**

1. What is the contribution of the subject alignment layer (Aₖ)? Have ablation experiments been conducted with its removal or freezing?

2. What are the individual contributions of the contrastive learning objective versus the subject alignment module (Aₖ)?  Have ablation experiments been performed to evaluate them?

3. The impacts of hyperparameters such as temperature parameter, negative sampling strategy, and batch size on the results have not been systematically explored, which limits the interpretability and generalizability of the method.

**Questions:**

see Weakness Section

---

> ### Author Response · Authors · 2025-11-21
>
> We thank the reviewer for their insightful comments. Below we address each point in detail.
>
> **Regarding Weakness 1**: The subject-alignment layer Ak is used to reduce inter-subject variability and enable cross-subject training within a shared embedding space. Its role is analogous to functional alignment methods used in recent large-scale cross-subject fMRI decoding frameworks such as TRIBE model (d’Ascoli et al. 2025), Tang et al. 2025 (“Semantic language decoding across participants and stimulus modalities”), and Ferrante et al. 2024 (for vision). These works demonstrate that anatomical alignment alone is insufficient for high-level decoding across subjects, and that subject-specific linear alignment significantly improves generalization.
> In our setting, the alignment layer is not a separate preprocessing step but is jointly optimized inside the contrastive training objective. This allows the model to learn a set of subject-specific linear transforms Ak that map each subject’s fMRI responses into a common functional space that maximizes contrastive similarity with the shared stimulus embedding. To address the reviewer’s question on ablation: removing the alignment layer corresponds to training a separate model for each subject (i.e., anatomical-only alignment). You can find the results in the following point and Appendix section A.3.
>
> **Regarding Weakness 2**: We thank the reviewer for raising this important question. Fully disentangling the contributions of (i) the subject-alignment module 𝐴k and (ii) the contrastive learning objective is indeed a valuable direction. In the current submission, we systematically ablate these two components separately:
>
> 1.	Effect of subject alignment.
> We compare the linear CL model (with 𝐴k) to an anatomical-only variant in which each subject is trained independently, removing cross-subject alignment. Results reported below show a consistent drop in performance across all datasets when alignment is removed.
>
> | Dset Method | Top1 | Top3 |
> |------|-----------|-----------|
> | NSD Anatomical CL | 17.8±1.11 | 34.5±1.49 |
> | HUTH Anatomical CL | 37.2±2.55 | 61.9±2.81 |
> | GTZAN Anatomical CL | 27.4±1.69 | 50.5±1.92 |
>
> 2.	Effect of the contrastive objective (linear CL vs Ridge).
> Independently, we compare the contrastive linear model to ridge regression, which uses a non-contrastive objective. As shown in Table 1, replacing the contrastive loss with ridge results in a substantial performance drop.
>
> Together, these two evaluations indicate that both the contrastive objective and the alignment module contribute meaningfully to the final performance.
>
> **Regarding Weakness 3**:
> Thanks for the comment. We agree that hyperparameters are important in contrastive learning. We use the standard positive/negative sampling of the NT-Xent loss (one positive per anchor and all other items in the batch as negatives). This is the same strategy used in SimCLR, CLIP, and recent contrastive brain–embedding models. Because this sampling is deterministic given the batch, it does not introduce instability or additional hyperparameters.
> Regarding temperature, we performed a dedicated grid search (τ ∈ {0.02, 0.05, 0.10, 0.50}) for each dataset. Performance varied smoothly across this range: moving one step away from the optimal τ typically changed Top-1 accuracy by 2-3 percentage points, suggesting that the decoder is reasonably robust to temperature choice. The values selected in the manuscript correspond to the best-performing τ per dataset.
> We tested multiple batch sizes (128, 512, 1024, 2048). The best performance consistently emerged with larger batch sizes (1024–2048). This is consistent with the behavior of contrastive losses: larger batches yield a larger number of in-batch negatives, which improves the estimation of the objective and stabilizes training. Smaller batches underperform due to fewer negatives and higher variance in the gradient.

---

### Official Review · Reviewer_iD6T · 2025-11-01

**Soundness:** 3
**Presentation:** 3
**Contribution:** 3
**Rating:** 6
**Confidence:** 3

**Summary:**

This paper investigates brain decoding from functional MRI (fMRI) across three modalities: images, text, and music. The authors propose that simple linear mappings trained with contrastive learning objectives consistently outperform both ridge regression and non-linear alternatives (shallow MLPs) when mapping brain activity to foundation model embedding spaces (CLIP for images, CLAP for audio, LLaMA for text). The core finding is that a subject-aligned linear transformation trained with NT-Xent contrastive loss achieves 10-15% improvements in Top-1 and Top-3 retrieval accuracy compared to baselines. The authors argue this success stems from: (1) concepts being organized as vectors in the brain where geometric relationships encode meaning, making contrastive objectives biologically plausible, and (2) fMRI measurements effectively linearizing neural dynamics through spatial/temporal averaging and noise, making linear models sufficient at the macroscale.

**Strengths:**

1. The consistent 10-15% improvement over ridge regression across all three modalities is a clear result (though statistical testing is needed).

2. Using the same architecture for images, text, and music is elegant and demonstrates generality. This is the paper's strongest contribution.

3. Acknowledging that non-linear architecture space is large and that more exploration could find better models is commendable.

**Weaknesses:**

1. Testing only shallow MLPs (≤5 layers, only ReLU/GELU) is insufficient to claim non-linear models fail. Modern architectures are completely unexplored:

2. Table 1 shows means ± std but no significance tests between ridge and linear CL. Are the improvements statistically significant given the sample sizes? What is the effect size?

**Questions:**

1. Why only test shallow MLPs with simple activations? Modern fMRI decoding papers use much more sophisticated architectures. Can you compare against MindEye2 (Scotti et al. 2024) or UMBRAE (Xia et al. 2024b) which use non-linear models successfully?

---

> ### Author Response · Authors · 2025-11-21
>
> We want to thank the reviewer for insightful comments. We reply point-to-point to each weakness/question raised.
>
> **Regarding weakness 1**:
> We thank the reviewer for highlighting that our original wording could be interpreted as an overgeneral statement about the failure of non-linear models. We have adjusted the text to clarify our scope. Specifically, our claim refers to the family of MLPs we evaluated; within this class, increasing non-linear capacity did not provide measurable benefits under the same data regime, training objective, and embedding space.
> To further strengthen this point, we extended our experiments to include a residual MLP with skip connections—one of the key components of modern architectures. Across all datasets, this model still underperformed the linear contrastive decoder. Quantitatively, the residual MLP shows a drop in Top-1 accuracy compared to the linear CL model (Table1), with paired t-tests confirming that these differences are statistically significant (p < 1e-8 in all datasets).
>
> | Dset | Top-1 | Top-3 |
> |------|-------|--------|
> | NSD | 15.88±4.42 | 32.28±3.10 |
> | HUTH | 34.35±3.93 | 55.19±4.52 |
> | GTZAN | 25.58±3.91 | 51.45±5.03 |
>
> Importantly, we do not claim that no non-linear architecture could ever outperform a linear mapping. Rather, our evidence shows that in the current fMRI preprocessing regime—which collapses temporal dynamics into GLM betas (NSD) or HRF-averaged responses (HUTH, GTZAN)—additional non-linear capacity appears prone to noise amplification. Because the temporal structure has been removed by design, architectures specialized for sequence modeling (e.g., LSTMs, Transformers) are not straightforwardly applicable without redesigning the entire pipeline. Investigating such architectures on minimally-averaged or single-TR fMRI is indeed an interesting direction, but it is orthogonal to the question we study here, which is: given the standard fMRI decoding setup, does added non-linear capacity improve retrieval accuracy?
>
> **Regarding weakness 2**:
> Thanks for pointing this out. For each dataset, we computed a paired t-test across subjects and seeds comparing Linear-CL vs Ridge and Linear-CL vs MLP. Effect sizes are reported using paired Cohen’s d. Across all datasets and metrics, the Linear-CL decoder shows statistically significant improvements (see Appendix A.1):
>
> | Dset | Compare | Top1 (t-stat,p-val,effect) | Top3 (t-stat,p-val,effect) |
> |------|------|-------------|-------------|
> | NSD | Linear-CL vs Ridge | 14.25, 1.3e-11, 2.59 | 17.78, 2.6e-13, 2.98 |
> | NSD | Linear-CL vs MLP | 19.03, 7.8e-14, 3.26 | 17.97, 2.2e-13, 3.02 |
> | HUTH | Linear-CL vs Ridge | 5.11, 4.4e-4, 1.23 | 6.78, 2.6e-5, 1.40 |
> | HUTH | Linear-CL vs MLP | 2.21, 4.4e-2, 0.59 | 3.70, 2.3e-3, 0.92 |
> | GTZAN | Linear-CL vs Ridge | 7.74, 5.2e-8, 1.55 | 12.96, 5.1e-13, 2.79 |
> | GTZAN | Linear-CL vs MLP | 8.36, 1.4e-8, 1.67 | 6.43, 1.2e-6, 1.28 |
>
> **Regarding question 1**:
> Thank you for the suggestion. We agree that recent fMRI–vision works such as MindEye2 (Scotti et al., 2024) and UMBRAE (Xia et al., 2024b) represent important progress in non-linear generative decoding. However, these models differ fundamentally from our setting, making a direct comparison non-trivial. First, our method performs semantic retrieval rather than image generation. MindEye2 and UMBRAE produce pixel-level reconstructions and are evaluated on reconstruction-oriented metrics. Second, MindEye2 and UMBRAE involve large, deep generative pipelines (diffusion models, multi-stage refinement), whereas our approach intentionally uses a lightweight linear projection model without a generative backbone. The goal of our paper is to analyze how much performance can be achieved purely from representation alignment.
> Nevertheless, to assess the compatibility of evaluation metrics, we computed low-level and high-level similarity scores between the ground-truth test images and the images retrieved (in the NSD framework) by our model. Despite the conceptual mismatch with reconstruction metrics, the values are competitive relative to the ranges reported in MindEye2/UMBRAE (see Appendix A.4):
>
> | Metric | Mean±Std |
> |--------|-----------|
> | AlexNet(2) | 0.9408±0.0649 |
> | AlexNet(5) | 0.8926±0.0869 |
> | CLIP-sim | 0.9163±0.1244 |
> | SSIM | 0.4993±0.3340 |
> | PixCorr | 0.3540±0.2369 |
> | EffNet-B1 dist | 0.6682±0.2397 |
>
> This confirms that retrieval-based decoding can reach competitive quality levels without requiring deep architectures. Finally, our intention is not to claim superiority over recent generative approaches, but rather to highlight how a simple linear architecture, grounded in brain–feature alignment, can already explain a large fraction of the decoding variance. This is consistent with prior evidence that, at fMRI resolution, neural responses generally behave in a linear regime.

---

### Author Response · Authors · 2025-11-21

We would like to sincerely thank the reviewers for their careful evaluation of our submission and for the constructive feedback that greatly helped us strengthen the paper. We have carefully addressed all raised concerns in our rebuttal and in the revised manuscript.

In particular, we:
(1) added the missing analyses requested by the reviewers, including MSE reporting, statistical significance tests, and ablations isolating the contributions of the subject-alignment layer;
(2) clarified the methodological details regarding voxel selection, retrieval-set size, hyperparameter exploration, and the role of the alignment module;
(3) expanded the discussion on non-linear architectures and included additional comparisons and similarity metrics to contextualize our results relative to recent brain decoding approaches;
(4) incorporated all suggested references and improved the clarity of the related-work section and main text.

All textual modifications in the main manuscript are marked in red in the uploaded PDF, while the entire Appendix—including new tables, expanded analyses, and additional ablation sections—is newly added in this revision.

---

### Note · Program_Chairs · 2026-01-17
**Submission Desk Rejected by Program Chairs**

The following references in this submission do not refer to real documents and/or have major errors in bibliographic information:

 Jack L. Gallant, Shinji Nishimoto, and Thomas Naselaris. The brain’s eye: Decoding mental images
from the human brain. Frontiers in Human Neuroscience, 6:68, 2012. doi: 10.3389/fnhum.2012.
00068